# Three-year functional, physical, and mental health outcomes after critical COVID-19: A prospective multicentre cohort study

Ingrid Didriksson[1,2�]*, Dorit Töniste[3,4he], Malin Hultgren[1‡], Martin Spångfors[1,5‡], Sara Göbel Andertun[6,7‡], Maria Nelderup[6,7‡], Anton Reepalu[8,9he], Attila Frigyesi[1,10he], Hans Friberg[1,2he], Gisela Lilja[3,4he]

1 Department of Clinical Sciences, Anaesthesiology and Intensive Care, Lund University, Lund, Sweden, 2 Department of Intensive and Perioperative Care, Skåne University Hospital, Malmö, Sweden, 3 Department of Clinical Sciences, Neurology, Lund, Lund University, Lund, Sweden, 4 Department of Neurology, Skåne University Hospital, Lund, Sweden, 5 Department of Anaesthesia and Intensive Care, Skåne University Hospital, Kristianstad, Sweden, 6 Department of Clinical Sciences, Lund University, Helsingborg, Sweden, 7 Department of Anaesthesia and Intensive Care, Helsingborg Hospital, Helsingborg, Sweden, 8 Department of Translational Medicine, Lund University, Malmö, Sweden, 9 Department of Infectious Diseases, Skåne University Hospital, Malmö, Sweden, 10 Department of Intensive and Perioperative Care, Skåne University Hospital, Lund, Sweden

☝ These authors contributed equally to this work.
‡ These authors also contributed equally to this work.
* ingrid_kristina_larsdotter.didriksson@med.lu.se

## Abstract

### Background

The understanding of recovery after critical COVID-19 beyond the first year is limited.

### Objectives

To describe changes in functional, physical, and mental health outcomes between 1 and 3 years among survivors of critical COVID-19 and to identify factors associated with incomplete recovery at 3 years.

### Methods

A prospective multicentre cohort study of survivors of critical COVID-19 with follow-up at 1 and 3 years. The primary outcome was functional outcome, assessed using the Glasgow Outcome Scale-Extended (GOSE), which ranges from 1 to 8, with scores of 6 or less indicating incomplete recovery. Secondary outcomes included return-to-work, physical and mental Health-Related Quality of Life (HRQoL), life satisfaction, fatigue, psychological symptoms (anxiety, depression, post-traumatic stress disorder), and respiratory symptoms. Multivariable logistic regression was used to identify factors associated with incomplete recovery (GOSE ≤ 6) at 3 years.

**Data availability statement:** The data underlying the findings of this study contain sensitive personal information and cannot be made publicly available under Swedish law (GDPR). The data are owned and managed by Region Skåne. Qualified researchers who meet the criteria for access to confidential data may request access through the Region Skåne Research Data Office via email (forskningsdata@skane.se). Requests will be assessed in accordance with institutional and ethical regulations. The authors do not have special access privileges.Requests will be assessed in accordance with institutional and ethical regulations. The authors do not have special access privileges.

**Funding:** This work received funding from the Swedish Heart-Lung Foundation (grant number 21004021; HF), the Swedish Research Council Formas (grant number 2019:YF0053; AF), and the Hans-Gabriel and Alice Trolle-Wachtmeister Foundation for Medical Research (HF). The funders had no role in study design, data collection and analysis, decision to publish, or preparation of the manuscript.

**Competing interests:** The authors have declared that no competing interests exist.

**List of abbreviations:** ARDS: Acute Respiratory Distress Syndrome; COVID-19: Coronavirus Disease 2019; GOSE: Glasgow Outcome Scale-Extended; HADS: Hospital Anxiety and Depression Scale; HADS-A: Hospital Anxiety and Depression Scale- Anxiety; HADS-D: Hospital Anxiety and Depression Scale- Depression; HRQoL: Health-Related Quality of Life; ICU: Intensive Care Unit; ICF: International Classification of Functioning, Disability and Health; MCS: Mental Component Summary; MFIS: Modified Fatigue Impact Scale; MID: Minimally Important Difference; OECD: Organisation for Economic Co-operation and Development; PCS: Physical Component Summary; PCL-5: PTSD Checklist for DSM-5; PICS: Post Intensive Care Syndrome; SARS-CoV-2: Severe Acute Respiratory Syndrome Coronavirus 2; SGRQ: St. George's Respiratory Questionnaire;SF-36v2®: Short Form Health Survey version 2; STROBE: Strengthening the reporting of observational studies in epidemiology; WHO: World Health Organisation.

## Results

Among 191 of 210 eligible participants, functional outcome declined from 1 to 3 years, and participants with incomplete recovery increased from 32% to 45%. Worse outcomes were observed in mental HRQoL, fatigue, depression, and post-traumatic stress, while return-to-work rates, physical HRQoL, life satisfaction, anxiety, and respiratory symptoms remained stable. Younger age [OR 0.70 (95% CI 0.54–0.91), $p = 0.008$] and higher Clinical Frailty Scale score [OR 1.54 (95% CI 1.04–2.28), $p = 0.029$] were independently associated with incomplete recovery at 3 years.

## Conclusions

Survivors of critical COVID-19 experienced a decline in functional outcome and worsening mental health between 1 and 3 years after ICU admission. Younger and frail survivors may require increased attention and support.

## Trial registration

ClinicalTrials.gov Identifier: NCT04974775, registered April 28, 2020.

## Introduction

### Background

The COVID-19 pandemic led to an unprecedented demand for critical care, mostly due to severe acute respiratory failure [1]. Survivors of COVID-19 are at risk of experiencing severe long-term health effects [2]. The WHO defines a post-COVID-19 condition that reflects the multi-organ nature and diverse clinical manifestations associated with COVID-19 [3], characterised by respiratory symptoms, fatigue, cognitive dysfunction, and an overall impact on everyday functioning [4].

Survivors of critical COVID-19 also share key physiological and functional features with the non-COVID Acute Respiratory Distress Syndrome (ARDS) [5], where studies of non-COVID ARDS survivors have shown substantial recovery within the first year, but with some impairments remaining [6,7]. These impairments often manifest as a Post-Intensive Care Syndrome (PICS), characterised by physical, cognitive, and mental health problems [8], highlighting the importance of structured long-term follow-up after critical illness [9]. Survivors of critical COVID-19 may thus have an increased risk for long-term effects influenced by both post-COVID-19 condition and PICS.

We previously reported improvements in functional outcome and physical Health-Related Quality of Life (HRQoL) between 3 months and 1 year among survivors of critical COVID-19. Despite this progress, physical HRQoL remained below normal levels, and many survivors experienced continued fatigue at 1 year [10,11]. While most studies have focused on outcomes within the first year [12–14], recent findings indicate that some symptoms persist well beyond this period [15]. Persistent physical and worsening mental symptoms have been reported in survivors of critical

COVID-19 at two years [16]. Other studies have identified female sex, acute disease severity, comorbidities, and invasive mechanical ventilation as risk factors for reduced HRQoL within two years after critical COVID-19 [13,15,17]. Recovery trajectories after critical COVID-19 should therefore be interpreted separately from those reported in general hospitalised COVID-19 populations. The CO-FLOW study found that 73% of hospitalised COVID-19 survivors had not fully recovered after two years, with ICU-treated patients demonstrating a slower recovery than non-ICU patients [18].

Long-term studies to date differ markedly in cohort size, follow-up duration, measurement tools, and adjustment for frailty or other baseline characteristics. These methodological variations make cross-study comparisons challenging and underscore the importance of longitudinal cohorts using consistent, validated outcome measures across multiple time points.

Against this background, knowledge of recovery after critical illness and post-COVID-19 condition suggests that long-term outcomes beyond the first year may be increasingly shaped by persistent post-COVID-19–related mechanisms and individual vulnerability, with age, frailty, comorbidities, and socioeconomic factors playing a more prominent role over time [3,19,20]. However, very few studies extend beyond 24 months, leaving long-term recovery trajectories after critical COVID-19 insufficiently investigated, including characteristics associated with sustained or delayed recovery.

### Objectives

To describe changes in functional, physical, and mental health outcomes between 1 and 3 years among survivors of critical COVID-19 and to identify factors associated with incomplete recovery at 3 years.

## Materials and methods

### Study design, setting, and participants

SWECRIT COVID-19, a prospective, observational, multicentre cohort study, included all critically ill patients (≥ 18 years) with laboratory-confirmed SARS-CoV-2 infection across six intensive care units (ICUs) in southern Sweden from May 11, 2020, to May 10, 2021 [21]. Patients whose primary reason for ICU admission was not COVID-19 were excluded from the study.

Surviving participants were invited to a structured follow-up at 3 months, 1 year, and 3 years after ICU admission. Eligibility for each time point required that participants were alive and had completed the preceding assessment. As this was a population-based cohort study with a predetermined recruitment period of 1 year, no formal sample size calculation was performed. This manuscript adheres to the STROBE guidelines for observational studies [22].

### Data collection

Methods for data collection during ICU care have been published previously [21]. Follow-up at 1 year was predominantly face-to-face [11], while the 3-year follow-up was conducted exclusively via telephone. This approach was chosen to ensure feasibility and to minimise loss to follow-up over the extended study period, allowing participation irrespective of geographical distance and reducing the burden associated with in-person visits. Questionnaires were sent to participants by postal mail for completion before the follow-ups. Certified interpreters were used whenever a presumed language barrier was present. Outcome assessors, including occupational therapists, physiotherapists, and ICU nurses, received training before their first follow-up. Participants in need of additional support were offered referrals in accordance with the principles of Good Clinical Practice.

### Ethical considerations

The Swedish Ethical Review Authority approved the study (ref. nos. 2020–01955, 2020–03483, 2021–00655), and written informed consent was obtained from participants during their hospital stay or at the follow-up visit, with the option to opt out at any time. Additional consent for the 3-year follow-up was obtained at the 1-year follow-up (ref. no. 2021–02323).

## Outcomes and outcome measures

The primary outcome of this study was functional outcome. Secondary outcomes were return-to-work, HRQoL, life satisfaction, fatigue, psychological symptoms (anxiety, depression, post-traumatic stress disorder), and respiratory symptoms.

**Functional outcome.** We assessed functional outcome using the Glasgow Outcome Scale-Extended (GOSE) [23], a clinician-reported ordinal scale ranging from 1 (death) to 8 (full recovery). We defined incomplete recovery as a GOSE score of 6 or less. Although initially designed for traumatic brain injury, the GOSE covers domains relevant to critical illness recovery, such as societal participation, including the ability to work and independence in daily activities. While GOSE is not formally included in the ARDS Core Outcome Measure Set [24], it has been used in various studies of critical illness [25,26]. All GOSE assessments were conducted using the structured interview format and manualised scoring procedures, which were developed to improve inter-rater reliability and ensure consistency across assessors and modes of administration, including telephone-based interviews [27,28]. Telephone-based administration of the GOSE has demonstrated good agreement with face-to-face assessment when conducted using structured interviews [29].

**Return-to-work.** Occupational status was tracked at 1 and 3 years for individuals employed before study inclusion as an additional measure of functional recovery.

**HRQoL.** We assessed HRQoL using the patient-reported Short-Form Health Survey 36-item version 2 (SF-36v2˚) [30], which is recommended as a core outcome measure for ARDS survivors [24]. It comprises 36 items across eight health domains (Physical Functioning, Role Physical, Bodily Pain, General Health, Vitality, Social Functioning, Role-Emotional, and Mental Health). Domain scores are combined into two summary scores: the Physical Component Summary (PCS) and the Mental Component Summary (MCS), which reflect overall physical and mental HRQoL, respectively. Scores are presented as standardised T-scores, where 50 represents the population mean (2009 US sample), with a Standard Deviation (SD) of 10. Scores of 45 or less at the individual level and 47 or less at the group level indicate below-normal health. The PCS and MCS have strong validity for distinguishing between clinically significant groups [31]. Minimally important differences (MIDs) are established through anchor-based methodologies, as presented in the SF-36v2˚ manual [32].

**Life satisfaction.** Life satisfaction was assessed using the World Value Survey's Life Satisfaction question, which was answered on a Visual Analogue Scale ranging from 1 ("completely dissatisfied") to 10 ("completely satisfied") [33].

**Fatigue.** The patient-reported Modified Fatigue Impact Scale (MFIS) [34] assesses the impact of fatigue across three domains: physical, cognitive, and psychosocial. The MFIS uses 5-point scales (0–4), with total scores ranging from 0 to 84. Scores of 38 or higher indicate clinically significant fatigue. The MFIS has been validated in various populations [35], including post-COVID-19 [36]. A MID ≥ 4 was used to indicate a clinically relevant change [37].

**Psychological symptoms.** The patient-reported Hospital Anxiety and Depression Scale (HADS) [38], a core ARDS outcome measure [24], comprises 14 items in two domains: HADS-Anxiety and HADS-Depression. Each domain scores 0–21, with scores ≥ 8 indicating clinically significant symptoms of anxiety or depression. A MID of ≥ 1.9–2.5 was used to indicate a clinically relevant change [39].

Symptoms of PTSD were assessed using the patient-reported PTSD Checklist for DSM-5 (PCL-5) [40], a 20-item instrument, where each item is scored from 0 to 4, yielding a total score range of 0–80. A total score ≥ 33 indicates clinically significant PTSD symptoms [41]. A MID of ≥ 6 was used to indicate a clinically relevant change [42].

**Respiratory symptoms.** The St. George's Respiratory Questionnaire (SGRQ) is a 50-item patient-reported instrument that assesses respiratory symptoms across three domains: Symptoms, Activity, and Impacts. Scores range from 0 to 100, with higher scores indicating poorer functioning [43,44]. The normative value for the total score is 8.4 (SD 11.3) [45]. We defined clinically significant respiratory symptoms as total scores exceeding 1 SD above the normative mean (i.e., < 19.7).

## Statistical analysis

Categorical data are presented as frequencies (n) and percentages (%). Continuous variables are presented as medians with interquartile ranges (25th-75th percentiles) or means with 95% confidence intervals (CI). Both medians and means

are reported for ordinal patient-reported outcome measures to facilitate interpretation. Patient-reported outcomes were analysed using complete-case analysis, and scale scores were calculated according to their standard scoring manuals. No imputation of missing data was performed.

Differences between 1 and 3 years were evaluated using the Sign test for ordinal variables, the Wilcoxon signed-rank test for continuous variables, and the McNemar test for categorical variables. Changes were presented as absolute mean differences, alongside established MID values and visualised using paired box plots.

Two separate analyses were performed to identify factors associated with incomplete recovery at 3 years. The first analysis explored associations with baseline characteristics, and the second explored associations with patient-reported outcomes at 3 years.

For the baseline characteristics analysis, we employed a Purposeful variable selection methodology [46] to identify baseline factors associated with incomplete recovery at 3 years. Pre-COVID-19 characteristics and baseline ICU variables were initially examined. We excluded variables with fewer than 10 events or more than 30% missing values from subsequent analyses to prevent model instability and inflated standard errors. Variables with $p < 0.20$ in the univariable analyses were eligible for entry into the multivariable model. The multivariable logistic regression analysis was performed using a backward selection approach (removal criterion: $p > 0.10$). Model fit was evaluated using the Hosmer-Lemeshow goodness-of-fit test.

For the patient-reported outcomes analysis, univariable logistic regression was used to examine the association between incomplete recovery at 3 years and each patient-reported outcome. Multivariable regression was not performed due to expected and confirmed strong multicollinearity among patient-reported outcomes. Instead, we constructed a correlation network plot in R, using the igraph, ggraph and tidygraph packages [47–49] to visualise relationships between variables. Pairwise Spearman correlation coefficients were computed, retaining only strong correlations with absolute values greater than 0.4.

To explore heterogeneity in long-term outcomes, age-stratified analyses (<50, 50–65, ≥65 years) were performed at the 3-year follow-up. Functional outcome (GOSE) was compared using the Kruskal–Wallis test, and HRQoL outcomes (PCS, MCS) using one-way ANOVA.

Statistical significance was determined at $p < 0.05$. Statistical analyses were performed using the Statistical Package for the Social Sciences (SPSS) version 27 and the R Studio version 1.2.1335.

## Results

### Participant flow and characteristics

Overall mortality was 40% at 1 year (198/498) and remained 40% at 3 years (201/498). Among survivors, follow-up was completed by 217 of 259 eligible participants (84%) at 1 year and by 191 of 210 eligible participants (91%) at 3 years (Fig 1, Table 1). A descriptive comparison of participants retained and lost to follow-up between 1 and 3 years showed small group differences (S1 Table).

### Functional outcomes

The distribution of functional outcome shifted toward lower categories between 1 and 3 years (GOSE 7 [6 –8 ] vs 7 [6 –7 ], $p < 0.001$), and the proportion of participants with incomplete recovery (GOSE ≤ 6) increased from 32% to 45% (Fig 2, Table 2).

This change was driven by individual transitions from complete recovery or good recovery (GOSE 7–8) to moderate disability (GOSE 5–6) (Fig 3).

Return-to-work rates remained stable at 82% across time points (Fig 4, Fig 5).

Functional outcome at 3 years differed across age groups (<50, 50–65, ≥65 years; Kruskal–Wallis p = 0.036), with older survivors more frequently clustering in GOSE 7 and younger survivors showing greater dispersion. No corresponding differences were observed for PCS or MCS at 3 years (S2 Table).

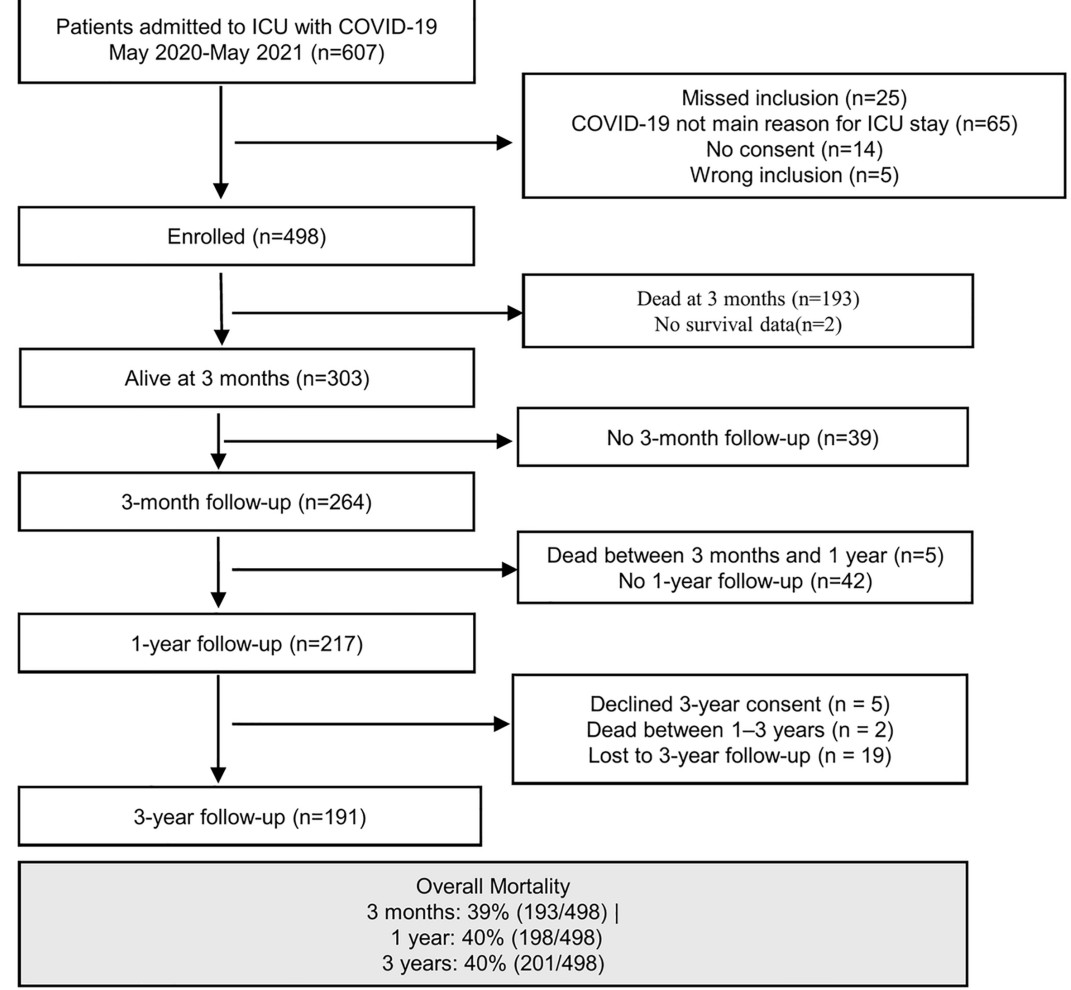

**Fig 1. Flow chart of recruitment and follow-up from ICU admission to 3-year assessments.** Overall mortality was 39% at 3 months (193/498), 40% at 1 year (198/498), and 40% at 3 years (201/498). Longitudinal follow-up was conducted among the 264 participants who completed the 3-month assessment. Of these, 259 (98%) were alive at 1 year, and 217 (84%) completed the 1-year follow-up. At 3 years, 191 (91%) of 210 eligible participants (5 declined consent, 2 died) completed the assessment, forming the final study cohort for longitudinal analysis.

## Health-related quality of life

Physical HRQoL (PCS) was unchanged, while mental HRQoL (MCS) deteriorated, exceeding the MID. The proportion of participants with scores below-normal mental HRQoL increased from 33% to 48%. Four SF-36v2® domains showed deterioration, but only Social Functioning and Mental Health exceeded MID thresholds. Life satisfaction remained unchanged (Table 2).

## Fatigue and psychological symptoms

Fatigue impact increased between 1 and 3 years, but did not meet the MID threshold, affecting 37% and 41% at respective timepoints (Table 2, S1 Table). Also, symptoms of depression and PTSD increased, though changes were below the MID threshold. Depressive symptoms affected 20% and 18% of participants at 1 and 3 years; PTSD affected 12% and 16%. Anxiety symptoms were stable over time, with approximately 25% affected at both timepoints (Table 2, Fig 6).

**Table 1. Patient demographics, admission variables, and ICU variables.**

| | All (including dead) | Survivors at follow-up | | |
| --- | --- | --- | --- | --- |
| | | 3 months | 1 year | 3 years |
| | (N = 498) | (n = 264) | (n = 217) | (n = 191) |
| **Demographics** | | | | |
| Age (years) | 66 [56-73] | 61 [52 –68] | 62 [53 –69 ] | 62 [53 –69 ] |
| Male | 74% | 73% | 73% | 73% |
| **Pre-COVID-19 characteristics** | | | | |
| BMI (kg/m²) | 30 [27 –35 ] | 31 [27 –36 ] | 31 [27 –35] | 31 [28 –35] |
| Smokers-ever | 44% | 37% | 38% | 39% |
| Charlson Comorbidity Index | 3 [2 –4 ] | 2 [1 –3] | 2 [1 –3] | 2 [1 –3] |
| Diabetes mellitus | 31% | 25% | 29% | 28% |
| Hypertension | 55% | 49% | 52% | 53% |
| Clinical Frailty Scale | 3 [2 –4] | 3 [2 –3 ] | 3 [2 –3] | 3 [2 –3] |
| **Socioeconomic factors** | | | | |
| Native Swedish speaker | N/A | 58% | 60% | 63% |
| Single household | N/A | 24% | 24% | 22% |
| Education > 12 years | N/A | 36% | 36% | 37% |
| Employed pre-COVID | N/A | 46% (121) | N/A | N/A |
| Employed pre-COVID age 24–65[a] | N/A | 63% (107) | N/A | N/A |
| **Admission parameters and intensive care variables** | | | | |
| SAPS3 | 60 [50-69] | 57 [47-66] | 57 [48-66] | 57 [48-66] |
| SOFA at admission | 8 [5 –9] | 7 [4 –9 ] | 8 [4 –9] | 8 [5 –9] |
| P/F ratio Day 1 min[b] | 9.0 [7.0-12] | 10 [7.0-13] | 10 [7.0-13] | 9.0 [7.0-12.3] |
| Severe ARDS P/F ratio < 13.3 kPa | 81% | 78% | 79% | 83% |
| PaCO2 Day 1 max[c] | 5.8 [4.9-7.4] | 5.7 [4.9-6.7] | 5.6 [4.9-6.7] | 5.5 [4.9-6.6] |
| IMV | 72% | 67% | 68% | 69% |
| IMV duration (days) | 9.8 [5.2-19] | 8.9 [4.5-18] | 9.0 [4.7-19] | 9 [5 –19] |
| Tracheostomy | 15% | 12% | 12% | 16% |
| CRRT | 14% | 13% | 13% | 17% |
| ICU stay (days) | 9.5 [4.9-17] | 10 [5.0-19] | 9.2 [4.7-18] | 10 [5 –19 ] |
| Hospital stay (days) | 23 [15-42] | 23 [14-45] | 23 [14-43] | 23 [15-45] |

Values are median [IQR] or % unless specified. [a]The denominator consists of participants aged 24–65 (n = 169). [b]Minimum P/F ratio during the first 24 hours of ICU admission. [c]Maximum PaCO2 during the first 24h of ICU admission.

BMI = Body Mass Index; SAPS3 = Simplified Acute Physiology Score 3; SOFA = Sequential Organ Failure Assessment; P/F = PaO2/FiO2; IMV = Invasive Mechanical Ventilation; CRRT = Continuous Renal Replacement Therapy.

## Respiratory symptoms

Respiratory symptoms did not differ materially, with 59% and 62% of participants showing clinically significant respiratory impairment at 1 and 3 years, respectively (Table 2, S3 Table, Fig 6).

## Factors associated with recovery

Participants with incomplete recovery at 3 years were younger and reported more symptoms in the patient-reported outcome measures compared to those with good recovery. Return-to-work rates were similar between groups; however, a higher proportion of participants with incomplete recovery were in rehabilitation and reported lower life satisfaction (Table 3). In multivariable logistic regression analysis, younger age (OR 0.70, 95% CI 0.54–0.91, $p = 0.008$, per 10-year increase) and

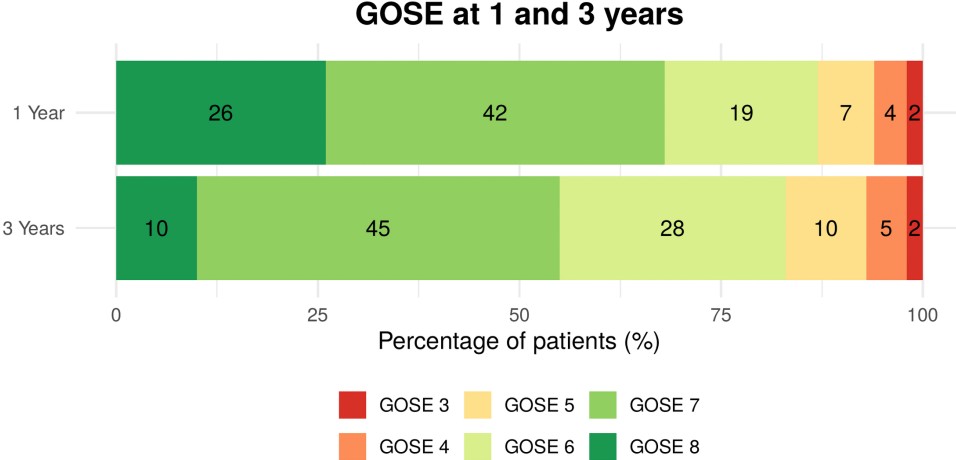

**Fig 2. Distribution of Glasgow Outcome Scale-Extended (GOSE) scores at 1-year and 3-year follow-up.** A horizontal bar chart showing the percentage distribution of GOSE scores among COVID-19 ICU survivors at the 1-year and 3-year follow-up. Each segment represents a specific GOSE score (3–8), with corresponding percentages labelled within each segment. GOSE 3–4 represents severe disability requiring assistance; GOSE 5–6 represents moderate disability with independent functioning but not a full return to pre-disease activities; GOSE 7–8 represents good recovery with minimal or no residual symptoms.

a higher pre-event Clinical Frailty Scale score (OR 1.54, 95% CI 1.04–2.28, $p = 0.029$) were independently associated with incomplete recovery at 3 years (Table 4). All patient-reported outcomes were significantly associated with incomplete recovery at 3 years in the univariable analyses (all $p < 0.001$) (S4 Table).

### Correlation analysis

The strongest correlation cluster among patient-reported outcomes (rs > 0.7) was observed among psychological measures: anxiety, depression, and PTSD symptoms. Fatigue (MFIS) showed strong correlations with both mental and physical measures (Fig 7, S5 Table).

### Discussion

The main findings of this study were that survivors of critical COVID-19 experienced a decline in functional outcome and mental health between the 1-year and 3-year follow-up. Close to half of these survivors reported problems with physical and mental HRQoL and fatigue at 3 years, while a majority experienced continued respiratory limitations. These findings highlight the substantial long-term health challenges faced by survivors of critical COVID-19.

Our previous work demonstrated improvements in functional outcome and physical HRQoL between 3 and 12 months post-ICU admission [10,11]. However, our 3-year data show a reversal of this positive trajectory, with more participants experiencing worsening health status and incomplete recovery, driven by a migration from complete recovery (GOSE 8) to moderate disability categories (GOSE 5–6). This non-linear recovery pattern diverges from traditional post-ICU recovery trajectories, in which continued improvement or stabilisation is typically seen [7].

While studies with follow-up extending to 2 years have generally described lingering but relatively stable symptoms after COVID-19 [16,18,50], our extended follow-up reveals mental health deterioration and declining mental HRQoL 3 years after critical COVID-19. These findings align with a recent population-based study [51] showing progressive worsening of psychiatric and cognitive symptoms 2–3 years after hospitalisation due to COVID-19.

Physical HRQoL showed no meaningful change from year 1 to year 3, but was consistently below normal health thresholds, and respiratory symptoms were a significant concern throughout the 3-year follow-up period, affecting

**Table 2. Outcomes at 1- and 3-year follow-up among survivors of critical COVID-19.**

| Outcome measure | | 1-year (n = 191) | 3-year (n = 191) | Mean difference and (MID) | p-value |
|---|---|---|---|---|---|
| **Functional outcome and return to work** | | | | | |
| GOSE | Functional outcome, median (IQR) | 7 [6 –8] | 7 [6 –7 ] | | < 0.001** |
| | Incomplete recovery (GOSE ≤ 6), % (n) | 32% (60) | 45% (84) | | |
| Return to work | Aged 24–65 years, % (n) | 82% (69/84) | 82% (69/84) | | |
| **HRQoL and Life Satisfaction** | | | | | |
| SF-36v2* | PCS, mean (95% CI) | 44.2 (42.7-45.7) | 44.8 (43.6-46.1) | 0.63 (2) | 0.140 |
| | Poor PCS (< 45), % | 53% | 50% | | |
| | MCS, mean (95% CI) | 48.9 (47.1-50.6) | 45.5 (43.6-47.3) | −3.39 (3)* | < 0.001** |
| | Poor MCS (< 45), % | 33% | 48% | | |
| | Life Satisfaction †VAS (95% CI) | 7.0 (6.2-7.3) | 6.8 (6.5-7.2) | | 0.640 |
| **Fatigue and psychological outcomes** | | | | | |
| MFIS | Fatigue (0–80), mean (95% CI) **Median (IQR)** | 31.6 (28.6-34.6) **31.0 [15.0-45.0]** | 34.6 (31.4-37.8) **35.0 [18.0-49.0]** | 3.0 (4.0) | 0.008** |
| | Fatigue above cut-off (≥ 38), % | 37% | 41% | | |
| HADS | Anxiety (0–20), mean (95% CI) **Median (IQR)** | 5.1 (4.4-5.7) **4.0 [2.0-8.0]** | 5.2 (4.4-6.0) **4.0 [1.0-8.0]** | −0.1 (2.0) | 0.540 |
| | Anxiety above cut-off (≥ 8), % | 25% | 24% | | |
| | Depression (0–20), mean (95% CI) **Median (IQR)** | 4.4 (3.8-5.0) **4.0 [1.0-7.0]** | 5.1 (4.4-5.8) **5.0 [1.5-8.0]** | 0.7 (2.5) | 0.006** |
| | Depression above cut-off (≥ 8), % | 20% | 18% | | |
| PCL-5 | Total score (0–80), mean (95% CI) **Median (IQR)** | 15.1 (12.7-17.5) **10.0 [4.0-21.0]** | 16.7 (13.9-19.4) **11.0 [4.0-24.0]** | 1.6 (6) | 0.041** |
| | Above cut-off (≥ 33), % | 12% | 16% | | |
| **Respiratory outcome** | | | | | |
| SGRQ | Total (0–100), mean (95% CI) **Median (IQR)** | 29.3 (25.8-32.9) **26.7 [9.4-46.0]** | 29.7 (26.1-33.4) **26.3 [11.2-41.4]** | 0.4 (4.0) | 0.350 |
| | ªSGRQ ≥ 19.7, % | 59% | 62% | | |

Number of responders: SF-36v2* (n = 146), Fatigue (n = 144), HADS (n = 145), PCL-5 (n = 145), SGRQ (n = 146).

Abbreviations: GOSE = Glasgow Outcome Scale-Extended; HRQoL = Health-Related Quality of Life; PCS = Physical Component Summary; MCS = Mental Component Summary; SF-36 = 36-Item Short Form Survey; HADS = Hospital Anxiety and Depression Scale; PCL-5 = PTSD Checklist for DSM-5; SGRQ = St. George's Respiratory Questionnaire; MID = Minimal Important Difference; CI = Confidence Interval; IQR = Interquartile Range. SGRQ norm mean = 8.4 (SD 11.3): Cut-off ≥ 19.7. †VAS = Visual Analogue Scale.

*above MID.

**statistically significant p < 0.05.

nearly two-thirds of participants. This plateau in physical health improvements after 1 year resembles patterns observed in non-COVID ARDS [6,7]. The concurrent deterioration in mental health, however, distinguishes critical COVID-19 from other forms of respiratory failure, where psychiatric symptoms typically stabilise after the first year [52]. The persistently high prevalence of fatigue also aligns with characteristics of the post-COVID condition [4,53].

Life satisfaction remained stable despite worsening functional outcome, with values nearing national averages [33]. This may reflect psychological adaptation [9] or a response shift following major illness [54].

Our correlation analysis identified two distinct symptom clusters, providing insights into potential treatment approaches. The strong interconnections between mental health measures (anxiety, depression, and PTSD symptoms) suggest shared underlying mechanisms, whereas functional outcome correlated more strongly with respiratory problems and fatigue.

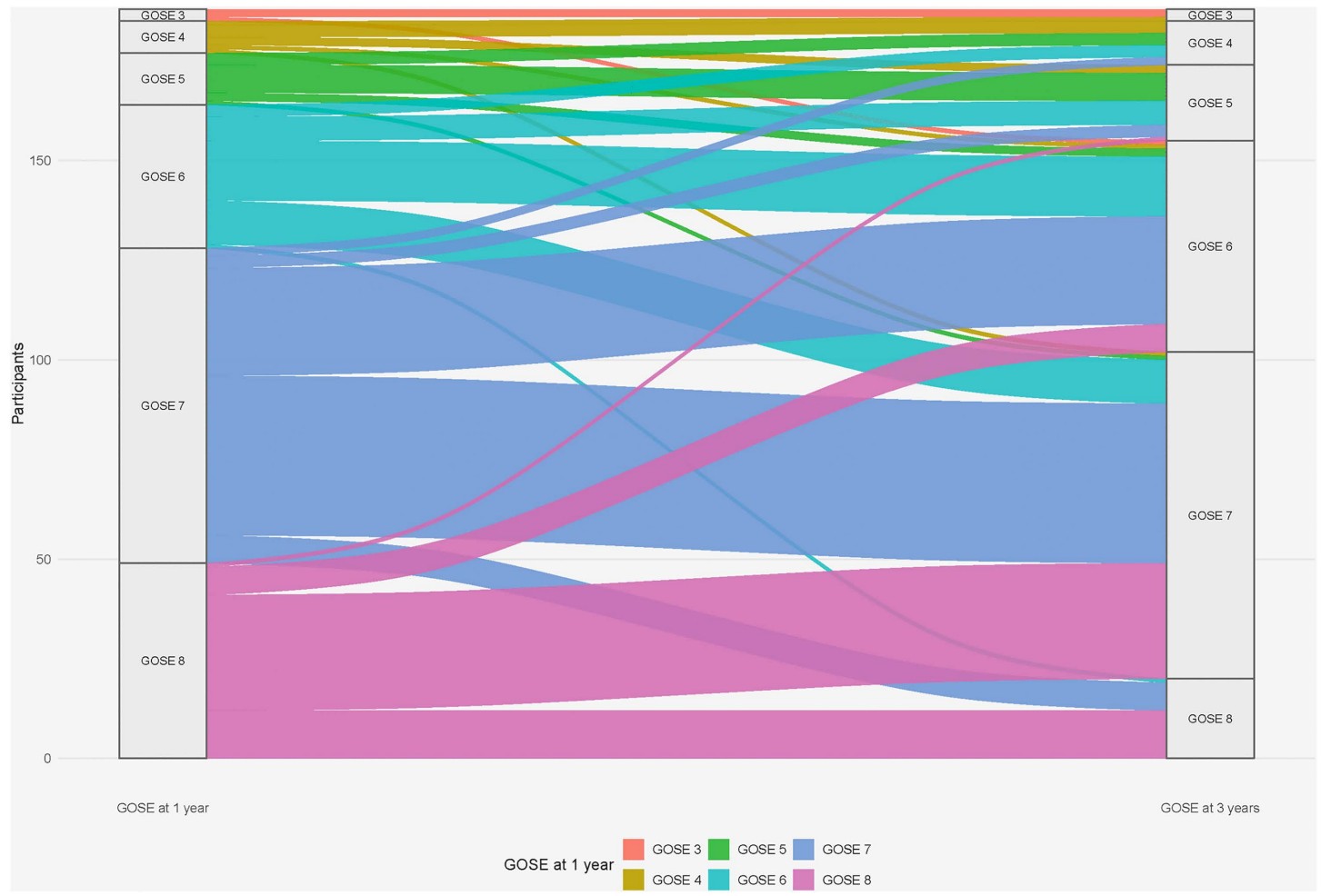

**Fig 3. Individual transitions in functional outcome between 1 and 3 years after critical COVID-19.** Alluvial plot illustrating individual changes in Glasgow Outcome Scale-Extended (GOSE) scores between the 1-year and 3-year follow-up among survivors of critical COVID-19 (n = 191). Each flow represents participants transitioning from a given GOSE category at 1 year (left) to their corresponding category at 3 years (right), with flow width proportional to the number of participants. The figure demonstrates a net shift from complete recovery (GOSE 8) and good recovery (GOSE 7) toward moderate disability (GOSE 5–6), while severe disability (GOSE 3–4) remained uncommon. GOSE = Glasgow Outcome Scale-Extended.

Fatigue, which captures physical, cognitive, and psychosocial impacts, may therefore serve as a bridge between symptom domains. These patterns suggest that integrated approaches addressing both physical and psychological symptoms may be more effective than treating each domain separately. This integrative perspective aligns with the WHO's International Classification of Functioning, Disability and Health (ICF) [55], which conceptualises functioning as arising from interactions between bodily functions, activity limitations, and participation.

The network-based patterns also highlight how the determinants of long-term outcome may shift over time. As in other long-term ICU follow-up studies, the influence of the initial critical illness appears to diminish over time [19,52]. In our cohort, this was reflected by the finding that baseline comorbidities and acute illness severity variables that predicted worse outcomes at 1 year, including diabetes mellitus and duration of mechanical ventilation, were not associated with incomplete recovery at 3 years. This was consistent with the network analysis, in which these factors showed weak or absent correlations with 3-year outcomes, suggesting decreasing relevance over time.

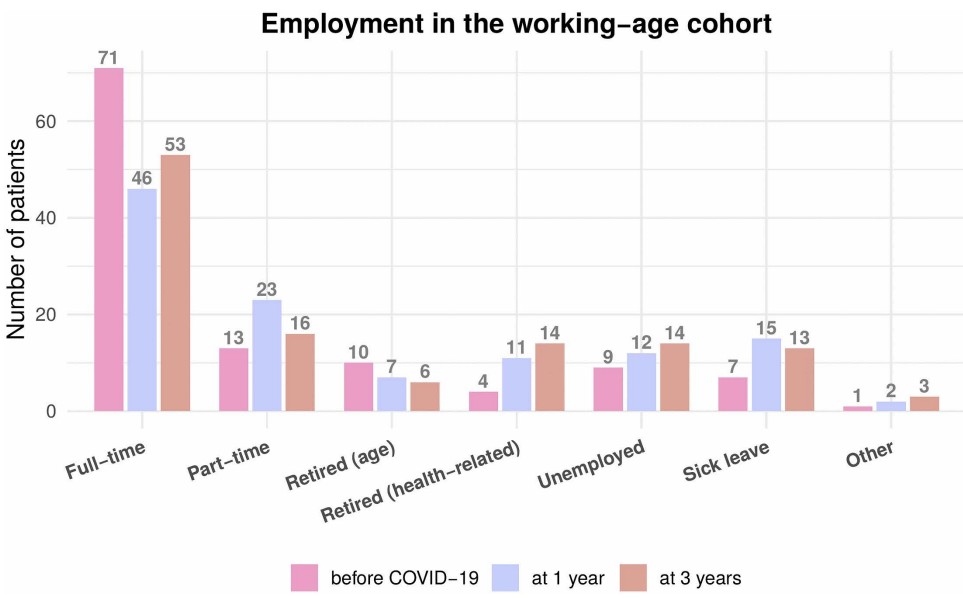

**Fig 4. Employment status and return-to-work rates in survivors of working age.** Employment distribution among the working-age cohort (24−65 years) at three time points: before COVID-19 (n = 84), and at 1- and 3-year follow-up. The grouped bars represent the number of participants in each employment category: full-time employed, part-time employed, retired due to age, retired due to sickness, unemployed, on sick leave, and others. Numbers within each segment represent the participants in that category at each time.

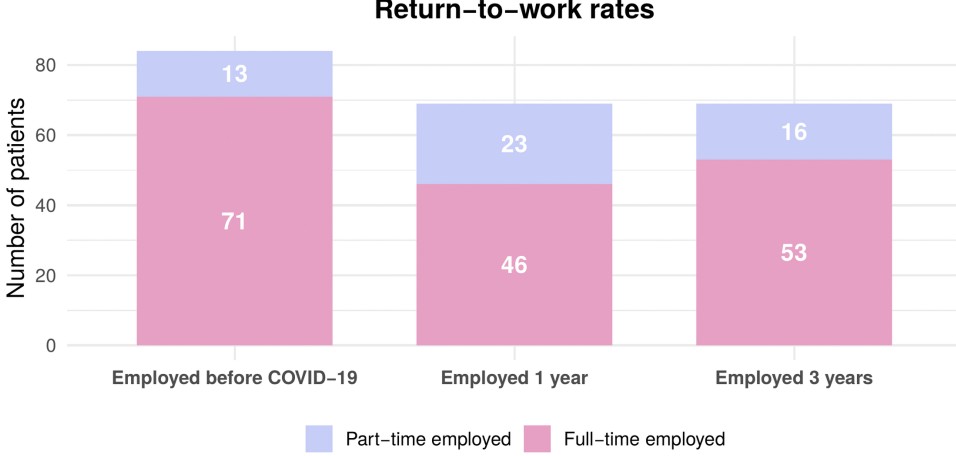

**Fig 5. Return-to-work rates among the working-age population employed before COVID-19 (n = 107).** The stacked bars illustrate the percentage of participants employed full-time (dark grey) versus part-time (light grey) at each time point: before the COVID-19 pandemic, as well as at 1- and 3-year follow-ups. Numbers within each segment represent the percentage of participants in that employment category.

In line with our previous study [11], age and frailty remained independently associated with long-term outcomes at 3 years. Counterintuitively, younger survivors had worse functional outcome than patients in older age groups, a pattern further supported by age-stratified analyses showing greater heterogeneity and a higher prevalence of incomplete recovery. From a biopsychosocial perspective [56], younger survivors may be more affected because they face greater role expectations and demands, whereas elderly patients, many of whom were around or above retirement age, often experience

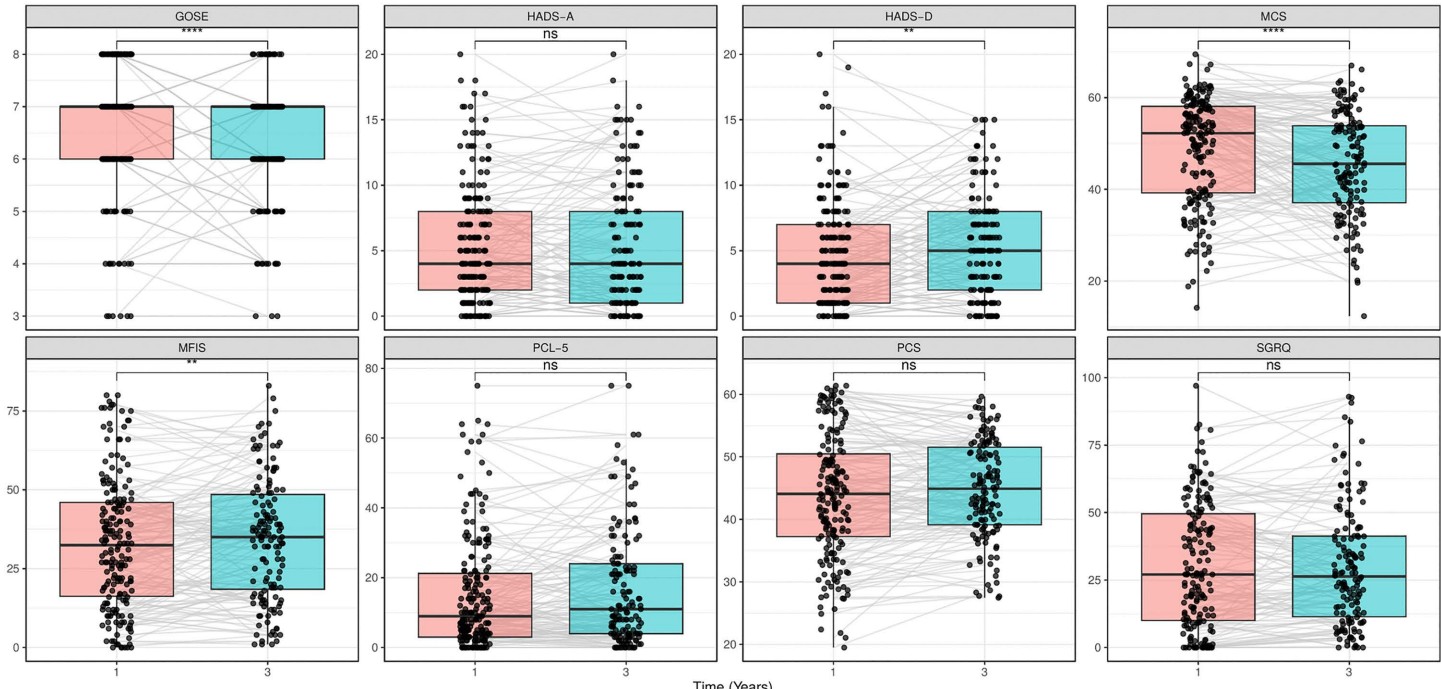

**Fig 6. Changes in Patient-Reported Outcomes from 1-year to 3-year follow-ups.** Box plots comparing patient-reported outcomes at 1-year (pink) and 3-year (blue) follow-ups. Each panel displays a different outcome measure: functional status (GOSE), anxiety (HADS-A), depression (HADS-D), mental HRQoL (MCS), fatigue (MFIS), post-traumatic stress (PCL-5), physical HRQoL (PCS), and respiratory symptoms (SGRQ). Boxes represent interquartile ranges with median lines; whiskers extend to values within 1.5 times the interquartile range (IQR), and dots show individual patient data. Grey lines connect measurements from the same individuals across time points. Statistical significance is indicated at the top of each panel (*p < 0.05, **p < 0.01, ***p < 0.001, ns = not significant). GOSE = Glasgow Outcome Scale-Extended; HADS-A/D = Hospital Anxiety and Depression Scale -Anxiety/ Depression; MCS/PCS = Mental/Physical Component Summary from SF-36v2˚; MFIS = Modified Fatigue Impact Scale; PCL-5 = Post-Traumatic Stress Disorder Checklist-5; SGRQ = St. George's Respiratory Questionnaire; HRQoL = Health-Related Quality of Life.

fewer such pressures [57,58]. Importantly, this pattern is not attributable to greater baseline vulnerability among younger patients, who in our cohort had lower frailty and fewer comorbidities at admission. A population-based study supports this pattern, showing that individuals younger than 50 had a higher risk of persistent post-COVID-19 symptoms than those aged 65 or older [20].

The post-viral immunological dysregulation theory [59,60] may provide a biological explanation for these age-related differences. This theory suggests that persistent immune abnormalities after SARS-CoV-2 can cause chronic inflammation, impairing physical health, worsening mental health, and increasing fatigue [61,62]. Younger individuals, who tend to mount stronger immune responses, may therefore be particularly vulnerable to prolonged immunological consequences.

The association between higher frailty and worse outcomes in the present study aligns with the COVIP findings, which showed that frailty was a stronger predictor of poor outcomes than age in elderly ICU patients with COVID-19 [63]. However, the proportion of frail participants in our cohort was lower than that of general ICU populations [64]. Several sociodemographic factors were included in the multivariable analyses but did not remain independently associated with incomplete recovery after multivariable adjustment, suggesting that long-term outcomes at three years were more strongly shaped by individual vulnerability than by sociodemographic characteristics. Together, these findings suggest that long-term recovery after critical COVID-19 is shaped more by post-acute factors and person-level vulnerabilities than by initial illness characteristics, contrasting with patterns reported previously [13,65].

**Table 3. Comparison of baseline demographics and patient-reported outcomes between participants with good and incomplete recovery at 3 years among survivors of critical COVID-19.**

| | Good Recovery (GOSE ≥ 7) (n = 104) | Incomplete Recovery (GOSE ≤ 6) (n = 86) | p-value |
|---|---|---|---|
| Age (years) | 65 [56 –70 ] | 58 [50 –65] | 0.002* |
| Male | 78% | 67% | 0.110 |
| BMI (kg/m²) | 30 [27 –34] | 32 [29 –36] | 0.023 |
| Clinical Frailty Scale | 3 [2 –3] | 3 [2 –3 ] | 0.210 |
| Life satisfaction | 7.6 (7.2-8.0) | 5.9 (5.3-6.5) | < 0.001* |
| PCS | 47 (46-49) | 42 (40-44) | < 0.001 |
| MCS | 52 (47-59) | 40 (38-43) | < 0.001 |
| HADS A (0–20) (n = 145) | 2.0 [1.0-5.0] | 7.0[3.0-11.0] | < 0.001 |
| HADS D (0–20) (n = 145) | 3.0[1.0-6.0] | 7.0 [4.0-9.0] | < 0.001 |
| PCL-5 total (0–80) (n = 145) | 6.0 [2.3-14.8] | 21.0 [9.0-34.5] | < 0.001 |
| Fatigue total (0–80) (n = 144) | 24.0 [12.3-39.0] | 42 [34.0-57.0] | < 0.001 |
| SGRQ Total (0–100) (n = 146) | 18.7 [9.1-33.2] | 36.5[24.0-54.9] | < 0.001 |
| Rehabilitation ongoing at 1 year | 10% | 23% | 0.011* |
| Rehabilitation ongoing at 3 years | 8% | 20% | 0.014* |
| Return-to-work | 46% | 54% | 0.087 |

Values: Median and IQR. †n = 145 for HADS A, HADS D, and PCL-5. BMI = Body Mass Index; GOSE = Glasgow Outcome Scale-Extended; HADS A = Hospital Anxiety and Depression Scale-Anxiety; HADS D = Depression; PCL-5 = PTSD Checklist-5; SGRQ = St. George's Respiratory Questionnaire.

*statistically significant p < 0.05

The absence of sex differences contrasts with other studies where females have shown worse recovery patterns [17]. However, the predominantly male cohort composition may have reduced our ability to detect such differences. The more frequent use of rehabilitation among participants with incomplete recovery may reflect greater perceived need rather than rehabilitation inefficacy [66].

Future research should investigate the underlying mechanisms linking the psychological symptom cluster (anxiety, depression, PTSD) with fatigue and evaluate integrated treatment approaches that address both physical and psychological domains simultaneously.

## Strengths and limitations

Strengths of this study include its size, the prospective multicentre design, and the extended 3-year follow-up. Using validated instruments for comprehensive outcome assessments enables meaningful comparisons over time. High retention between 1 and 3 years, together with similar baseline characteristics among retained and lost participants, reduced the risk of attrition bias (S1 Table).

This study also has several limitations.

First, the transition from face-to-face to telephone-based follow-up between years 1 and 3 may have introduced measurement-related limitations. Although telephone administration of the GOSE has shown good agreement with face-to-face assessment [29], validation specifically in post-COVID ICU populations is limited, and the absence of visual cues may influence clinician-reported outcomes.

Second, as this population-based cohort had a fixed recruitment period, no formal sample size calculation was performed. While the overall sample size was sufficient for the primary longitudinal analyses, statistical power for multivariable regression analyses may have been limited. Accordingly, null findings, including the lack of an independent association for sex, should be interpreted with caution.

**Table 4. Logistic regression analyses of baseline and ICU characteristics associated with incomplete recovery at 3 years (GOSE ≤ 6).**

| Univariable regression | | | Multivariable regression | | |
|---|---|---|---|---|---|
| *Variable* | *OR [95% CI]* | *p-value* | *Variable* | *OR [95% CI]* | *p-value* |
| Age (years/10) | 0.74 [0.58-0.94] | 0.016* | Age (years/10) | 0.70 [0.54-0.91] | 0.008* |
| Male | 0.59 [0.31-1.12] | 0.108 | Clinical Frailty Scale | 1.54 [1.04-2.28] | 0.029* |
| BMI (kg/m²) | 1.06 [1.01-1.12] | 0.029* | Native Swedish speaker | 0.57 [0.30-1.07] | 0.080 |
| Clinical Frailty Scale | 1.30 [0.91-1.87] | 0.146 | | | |
| Hypertension | 1.44 [0.81-2.57] | 0.214 | | | |
| Diabetes mellitus | 1.71 [0.69-4.28] | 0.250 | | | |
| COPD | 1.10 [0.53-2.28] | 0.790 | | | |
| Smoker | 1.03 [0.57-1.85] | 0.922 | | | |
| SAPS 3 | 1.00 [0.97-1.02] | 0.754 | | | |
| P/F ratio Day 1 min (kPa) | 1.01 [0.96-1.05] | 0.805 | | | |
| Tracheostomy | 1.00 [0.46-2.16] | 0.990 | | | |
| CRRT | 1.36 [0.64-2.87] | 0.428 | | | |
| IMV | 1.03 [0.55-1.91] | 0.936 | | | |
| $Log_{10}$ Duration of IMV | 1.23 [0.55-2.74] | 0.610 | | | |
| Hospital length of stay (days) | 1.01 [1.00-1.02] | 0.207 | | | |
| ICU Length of stay (days) | 1.01 [1.00-1.03] | 0.273 | | | |
| Single household | 1.47 [0.74-2.92] | 0.275 | | | |
| Native Swedish speaker | 0.64 [0.36-1.16] | 0.144 | | | |
| Level of education > 12 years | 1.52 [0.84-2.76] | 0.167 | | | |

OR = Odds Ratio; CI = Confidence Interval; GOSE = Glasgow Outcome Scale-Extended; BMI = Body Mass Index; COPD = Chronic Obstructive Pulmonary Disease; SAPS 3 = Simplified Acute Physiology Score 3; P/F = $PaO_2/FiO_2$; CRRT = Continuous Renal Replacement Therapy; IMV = Invasive Mechanical Ventilation; ICU = Intensive Care Unit. Age was modelled per 10-year increase in all logistic regression analyses.

*statistically significant p < 0.05.

Third, acute-phase mortality in our cohort was higher than in other Nordic cohorts (Chew et al., 2022), likely reflecting more severe respiratory failure at ICU admission. This may limit the generalisability of our findings to less severely ill populations.

Fourth, systematic psychiatric diagnoses were not collected at baseline, which limits the ability to provide a broader explanatory account of psychological outcomes and risk factors. We acknowledge that participants may have received more support than standard care, as symptoms were acknowledged during the 1-year follow-up. Limited information on rehabilitation or psychiatric interventions further limits our ability to evaluate their influence.

Fifth, using population norms rather than contemporary controls limits the ability to distinguish effects specific to critical COVID-19 from broader pandemic-related influences. Population norms mainly reflect pre-pandemic conditions, and population-level mental health changes during the pandemic were generally small, according to recent meta-analytic evidence [67]. Nevertheless, the longitudinal design, in which each participant serves as their own control, partially compensates for the lack of baseline psychiatric data and contemporary non-COVID controls by allowing recovery trajectories to be evaluated within individuals over time.

Finally, cultural and linguistic factors may have influenced patient-reported outcomes among non-native Swedish speakers, although the use of validated translations, interpreters, and US normative scoring for SF-36v2® reduces this risk.

## Conclusions

Three years after critical COVID-19, survivors showed worsening functional outcome and mental health compared to assessments at one year, while physical health was unchanged but remained impaired. These findings indicate a need

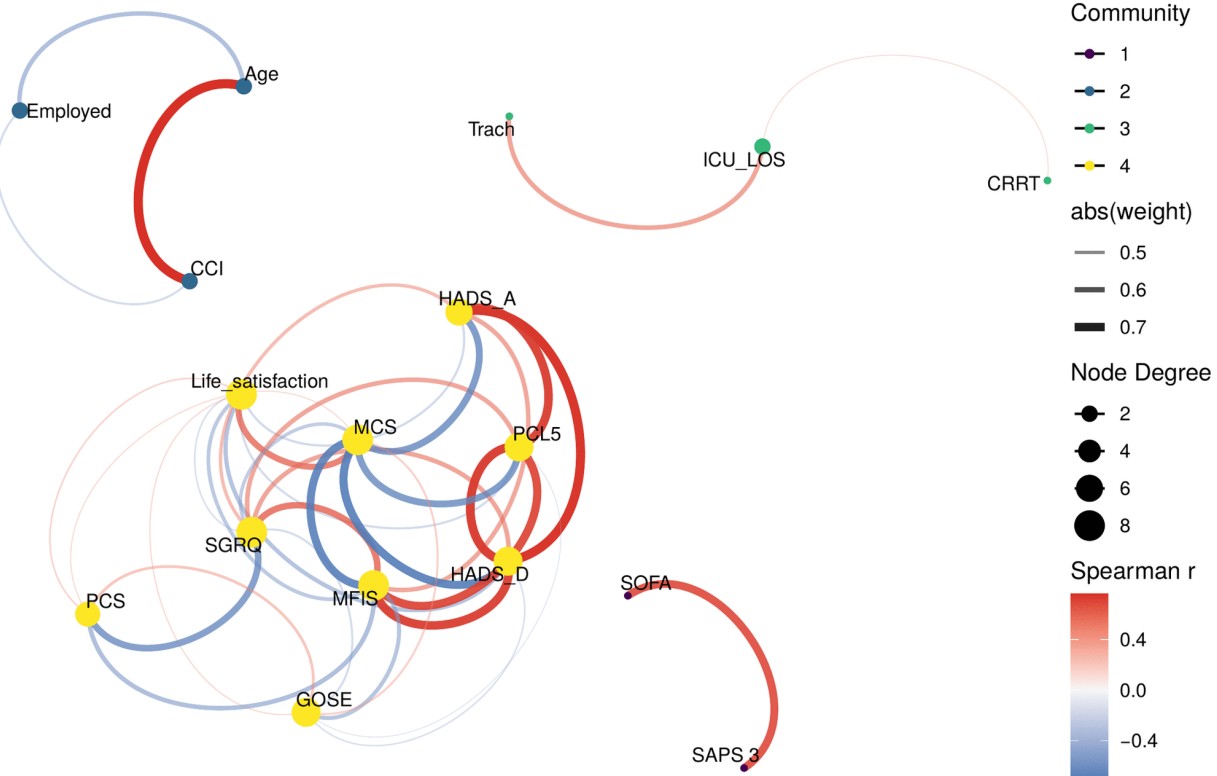

**Fig 7. Network correlation plot of variables in the COVID-19 ICU survivor cohort at 3-year follow-up.** Network plot showing correlations (|r| > 0.4) between demographic, clinical, and patient-reported outcomes at 3 years post-ICU admission. The edge colour indicates the direction of the correlation (red = positive and blue = negative), with thickness representing the strength of the correlation. The analysis reveals two main symptom clusters: (1) a tightly interconnected mental health cluster (left) comprising anxiety (HADS-A), depression (HADS-D), PTSD symptoms (PCL-5), and fatigue (MFIS), and (2) a physical-functional cluster linking respiratory symptoms (SGRQ), physical HRQoL (PCS), and functional outcome (GOSE). Mental HRQoL (MCS) bridges both clusters. GOSE = Glasgow Outcome Scale-Extended; HADS-A/D = Hospital Anxiety and Depression Scale -Anxiety/Depression; PCL-5 = Post-Traumatic Stress Disorder Checklist-5; MFIS = Modified Fatigue Impact Scale; SGRQ = St. George's Respiratory Questionnaire; PCS/MCS = Physical/Mental Component Summary from SF-36v2®; HRQoL = Health-Related Quality of Life.

for extended follow-up care beyond the first year post-ICU in critical COVID-19 disease. Younger patients and those with pre-existing frailty may require tailored support.

## Supporting information

**S1 Table. Comparison of participants retained in the 3-year follow-up and those lost between 1 and 3 years.** Descriptive comparison of demographic and clinical characteristics between retained and lost participants.
(DOCX)

**S2 Table. Functional outcome (GOSE) and health-related quality of life assessed by the Physical and Mental Component Summary scores (PCS and MCS) of the SF-36v2® at the 3-year follow-up, stratified by age (<50, 50–65, and ≥ 65 years).** GOSE was analysed using the Kruskal–Wallis test, and PCS and MCS using one-way analysis of variance.
(DOCX)

**S3 Table. Subdomains of patient-reported outcome measures at 1 and 3 years.** Descriptive results for all subdomains of SF-36v2, HADS, PCL-5, MFIS, and SGRQ at 1- and 3-year follow-up.
(DOCX)

**S4 Table. Univariable logistic regression analysis of patient-reported outcomes associated with incomplete recovery (GOSE ≤ 6) at 3 years.** Odds ratios, 95% confidence intervals, and p-values for each patient-reported outcome measure.
(DOCX)

**S5 Table. Spearman correlation coefficients among clinical outcome measures and patient-reported outcome measures at 3 years.** Correlation matrix showing relationships between GOSE, PCS, MCS, SGRQ, fatigue scores, and psychological symptom measures.
(DOCX)

## Acknowledgments

We thank all patients and their next of kin who participated in this study, as well as all staff at the ICUs of Skåne University Hospital in Malmö and Lund, Helsingborg Hospital, and Kristianstad Hospital. We also thank occupational therapists Anna Bjärnroos, Erik Mellerstedt, and Eva M. Johnsson, physiotherapist Katarina Heimburg, and research nurse Susann Schrey. Large language model assistance (Claude, Anthropic) was used exclusively for language editing; all content was reviewed and approved by the authors.

## Author contributions

**Conceptualization:** Hans Friberg, Gisela Lilja.

**Data curation:** Ingrid Didriksson, Dorit Töniste, Malin Hultgren, Martin Spångfors, Sara Göbel Andertun, Maria Nelderup, Anton Reepalu, Attila Frigyesi, Gisela Lilja.

**Formal analysis:** Ingrid Didriksson, Attila Frigyesi.

**Funding acquisition:** Ingrid Didriksson, Attila Frigyesi, Hans Friberg.

**Methodology:** Ingrid Didriksson, Attila Frigyesi, Hans Friberg, Gisela Lilja.

**Project administration:** Ingrid Didriksson, Martin Spångfors, Attila Frigyesi, Hans Friberg, Gisela Lilja.

**Resources:** Hans Friberg.

**Software:** Attila Frigyesi.

**Supervision:** Hans Friberg, Gisela Lilja.

**Validation:** Ingrid Didriksson, Dorit Töniste, Malin Hultgren, Martin Spångfors, Sara Göbel Andertun, Maria Nelderup, Anton Reepalu, Attila Frigyesi, Gisela Lilja.

**Visualization:** Ingrid Didriksson, Attila Frigyesi.

**Writing – original draft:** Ingrid Didriksson.

**Writing – review & editing:** Ingrid Didriksson, Dorit Töniste, Malin Hultgren, Martin Spångfors, Sara Göbel Andertun, Maria Nelderup, Anton Reepalu, Attila Frigyesi, Hans Friberg, Gisela Lilja.

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
