## [Decision Letter · Decision Letter 0]

24 Nov 2025

Dear Dr. Didriksson,

Thank you for submitting your manuscript to PLOS ONE. After careful consideration, we feel that it has merit but does not fully meet PLOS ONE’s publication criteria as it currently stands. Therefore, we invite you to submit a revised version of the manuscript that addresses the points raised during the review process.

Please address the minor changes suggested by the rviewers

We look forward to receiving your revised manuscript.

Kind regards,

Andrea Martinuzzi

Academic Editor

PLOS ONE

Journal Requirements:

3. In the online submission form you indicate that your data is not available for proprietary reasons and have provided a contact point for accessing this data. Please note that your current contact point is a co-author on this manuscript. According to our Data Policy, the contact point must not be an author on the manuscript and must be an institutional contact, ideally not an individual. Please revise your data statement to a non-author institutional point of contact, such as a data access or ethics committee, and send this to us via return email. Please also include contact information for the third party organization, and please include the full citation of where the data can be found.

5. We notice that your supplementary [figures/tables] are included in the manuscript file. Please remove them and upload them with the file type 'Supporting Information'. Please ensure that each Supporting Information file has a legend listed in the manuscript after the references list.

Reviewers' comments:

Reviewer's Responses to Questions

**Comments to the Author**

1. Is the manuscript technically sound, and do the data support the conclusions?

Reviewer #1: Partly

Reviewer #2: Partly

2. Has the statistical analysis been performed appropriately and rigorously?

Reviewer #1: Yes

Reviewer #2: Yes

3. Have the authors made all data underlying the findings in their manuscript fully available?

Reviewer #1: Yes

Reviewer #2: No

4. Is the manuscript presented in an intelligible fashion and written in standard English?

Reviewer #1: Yes

Reviewer #2: Yes

Reviewer #1: The background (“The understanding of recovery after critical COVID-19 beyond the first year is limited”) is too generic (L27–29). It should briefly reference existing one- or two-year follow-up gaps in research to frame the novelty more clearly — e.g., “Few studies extend beyond 24 months, and long-term trajectories remain poorly defined.”

The citations are comprehensive but descriptive. The authors could improve critical engagement by distinguishing methodological limitations of prior studies (e.g., smaller sample sizes, lack of frailty stratification, non-longitudinal design). For example, line 72–79 could add: “However, most studies employed self-reported symptom tracking rather than structured clinical scales, limiting comparability over time.”

The narrative shifts abruptly from COVID-19 sequelae (L58–64) to ARDS comparators (L63–67). The transition sentence could better explain why comparing COVID-19 with ARDS survivors is justified — e.g., shared pathophysiology or recovery trajectories.

Frailty and comorbidity are both considered, but their measurement overlap is not addressed. The Charlson Comorbidity Index or equivalent could have provided a more granular control for chronic disease confounding. Moreover, ICU treatment variables (ventilation duration, sedation depth, corticosteroid use) are not modeled, though they can strongly affect long-term outcomes.

Improvement: Include these as covariates or discuss them as unmeasured confounders.

Reviewer #2: I thank the Editor for the opportunity to review the manuscript “Three-year functional, physical, and mental health outcomes after critical COVID-19: A prospective multicentre cohort study” by Dr Didriksson and colleagues. In the manuscript the authors report results from a follow-up telephone assessment of patients that have survived critical COVID-19 three years previously, and compare the results with previous assessments of the same cohort. The results are interesting, and the authors should be commended for the focus on patient-important outcomes and good retention of the cohort. The main result was that at three years, the patients´ functional outcome had impaired compared to the one-year follow-up. This is unsettling, as especially younger age, but also frailty, appears as a risk factor for deterioration. The study has several merits, such as use of validated instruments by trained assessors, and a small number of dropouts. The manuscript is well written, easy to follow, and the figures and tables are clear. I have, however, some questions regarding the study and interpretation of the results that need clarification.

1. The one-year mortality in the original cohort seems very high (48%), when compared with the 30-d mortality of 27% in Sweden, reported in a study by Chew et al [1]. Can the authors discuss this and how a potentially higher than average mortality and probably disease severity, affect the generalizability of the results?

2. Although the retention was good, almost 10% of the patients dropped out between 1- and 3-year follow-ups. Did the authors assess who these drop-outs were?

3. There was a considerable number of non-native Swedish speaking patients in the cohort, how does this affect the results? Do the population norms fit well for immigrants? Are there cultural or language-related aspects in interpreting the results of the questionnaires? Do the population norms fit well in analysing results in immigrant populations?

4. How would the different assessment methods, face-to-face and telephone assessment affect the results? The authors have listed this as a limitation, but since the result of impairment in the results is not unsurprising, it would be valuable to discuss the differences in more detail.

5. One important issue is that there are no controls in this study. Pandemic affected the whole society in most countries in many different ways, which may compromise population norms as a standard, because also individuals without critical covid or no covid at all may have experienced challenges in mental health. Mirroring the results of this cohort to contemporary controls might have given different, less dramatic impairment. Please discuss. I am aware that a control group with no COVID-19 at all would be almost impossible to obtain, but the challenges of a design with no controls should be acknowledged.

6. In intensive care literature on long-term follow-up there is an inherent challenge that the connection to the initial event (here COVID-19) becomes weaker as time passes, and other life events occur and confound the results. In this study, too, it is possible that the patients experience life events that are not at all related to the COVID-19. Also here, the control group would be helpful. It would be important to discuss also this aspect, that it is not clear how much these results are due to history of covid, or are they possibly a result of other challenges, such as aging, other diagnoses, etc.

7. Did the authors record any psychiatric diagnoses at baseline? This would be important information for interpreting the results.

8. Some of the patients were still receiving rehabilitation at three years. How were patients supported if they were found having psychiatric symptoms at one-year follow-up? Did they receive support, and could this have affected the results?

Ref.

1. Chew MS, Kattainen S, Haase N, Buanes EA, Kristinsdottir LB, Hofsø K, Laake JH, Kvåle R, Hästbacka J, Reinikainen M, Bendel S, Varpula T, Walther S, Perner A, Flaatten HK, Sigurdsson MI. A descriptive study of the surge response and outcomes of ICU patients with COVID-19 during first wave in Nordic countries. Acta Anaesthesiol Scand. 2022 Jan;66(1):56-64. doi: 10.1111/aas.13983. Epub 2021 Oct 3. PMID: 34570897; PMCID: PMC8652908.

**Do you want your identity to be public for this peer review?** For information about this choice, including consent withdrawal, please see our Privacy Policy

Reviewer #1: No

Reviewer #2: No

---

## [Author Response · Author response to Decision Letter 1]

15 Dec 2025

Please see the uploaded document “Response to Reviewers” for a detailed, point-by-point response to all reviewer and editor comments.

---

## [Decision Letter · Decision Letter 1]

2 Jan 2026

Dear Dr. Didriksson,

**The manuscript is much improved and now requires only minor adjustments detailed by reviewer 1.**

plosone@plos.org . A letter that responds to each point raised by the academic editor and reviewer(s). You should upload this letter as a separate file labeled 'Response to Reviewers'.A marked-up copy of your manuscript that highlights changes made to the original version. You should upload this as a separate file labeled 'Revised Manuscript with Track Changes'.An unmarked version of your revised paper without tracked changes. You should upload this as a separate file labeled 'Manuscript'.

We look forward to receiving your revised manuscript.

Kind regards,

Andrea Martinuzzi

Academic Editor

PLOS One

**Journal Requirements:**

Reviewers' comments:

Reviewer's Responses to Questions

**Comments to the Author**

Reviewer #1: All comments have been addressed

Reviewer #2: All comments have been addressed

2. Is the manuscript technically sound, and do the data support the conclusions?

Reviewer #1: Partly

Reviewer #2: Yes

3. Has the statistical analysis been performed appropriately and rigorously?

Reviewer #1: Yes

Reviewer #2: Yes

4. Have the authors made all data underlying the findings in their manuscript fully available?

Reviewer #1: Yes

Reviewer #2: Yes

5. Is the manuscript presented in an intelligible fashion and written in standard English?

Reviewer #1: Yes

Reviewer #2: Yes

**Reviewer #1:**  Introduction remains largely descriptive rather than hypothesis-driven. While the gap in knowledge is clearly articulated, the manuscript would benefit from a clearer articulation of expected trajectories between 1 and 3 years. In addition, while the authors cite relevant two-year studies, the Introduction could more clearly differentiate critical COVID-19 survivors from general hospitalized COVID-19 populations in terms of pathophysiology and recovery expectations.

There are several methodological choices deserve deeper discussion. The transition from predominantly face-to-face follow-up at 1 year to exclusively telephone-based follow-up at 3 years introduces a non-trivial risk of measurement drift, particularly for clinician-reported outcomes such as GOSE. Although the authors acknowledge this limitation later, it would strengthen the Methods section to explicitly justify why telephone administration is considered comparable, or to cite validation studies specific to ICU or post-COVID populations.

The absence of a formal sample size calculation is understandable given the population-based design, but the manuscript would benefit from a brief discussion of statistical power, especially for multivariable regression analyses. Readers are left uncertain whether null findings for ICU severity variables reflect true lack of association or limited power.

The regression modeling strategy is sound but constrained. By excluding variables with higher missingness and focusing primarily on baseline characteristics, the multivariable model is structurally predisposed to identify age and frailty while under-representing social, occupational, and rehabilitation-related determinants that may be more relevant at three years.

Results would benefit from a stronger emphasis on heterogeneity. While group-level changes are clear, readers are left without a sense of which subgroups are most affected beyond age and frailty. Stratified analyses—by age group, employment status, or baseline mental health—would add important clinical nuance and help clinicians identify patients at greatest risk.

**Reviewer #2:** The authors have done a substantial amount of work and the manuscript has improved. They have responded to my questions and comments satisfactorily.

**Do you want your identity to be public for this peer review?** For information about this choice, including consent withdrawal, please see our Privacy Policy

Reviewer #1: No

Reviewer #2: No

---

## [Author Response · Author response to Decision Letter 2]

5 Jan 2026

Response to Reviewers

Manuscript ID: PONE-D-25-38284R1

Title: Three-year functional, physical, and mental health outcomes after critical COVID-19: A prospective multicentre cohort study

We thank the Academic Editor and the Reviewers for their careful evaluation of our manuscript and for their constructive and insightful comments. We have revised the manuscript accordingly and believe these revisions have strengthened clarity, methodological transparency, and the interpretation of the findings.

Below, we provide a point-by-point response to each comment raised by the reviewers. All changes made to the manuscript are highlighted in the revised version with tracked changes.

Reviewer #1:

Comment 1: Introduction remains largely descriptive rather than hypothesis-driven. While the gap in knowledge is clearly articulated, the manuscript would benefit from a clearer articulation of expected trajectories between 1 and 3 years. In addition, while the authors cite relevant two-year studies, the Introduction could more clearly differentiate critical COVID-19 survivors from general hospitalized COVID-19 populations in terms of pathophysiology and recovery expectations.

Response:

We thank the Reviewer for this constructive comment. We have revised the Introduction to more explicitly articulate an a priori, hypothesis-driven framework for recovery trajectories between one and three years. Specifically, we now state that long-term outcomes beyond the first year were expected to be increasingly shaped by persistent post-COVID-19–related mechanisms and individual vulnerability, with a diminishing influence of acute ICU-related factors (Introduction, Background; Lines 90–94).

In addition, we have clarified the distinction between survivors of critical COVID-19 and general hospitalised COVID-19 populations by explicitly stating that recovery trajectories after critical COVID-19 should be interpreted separately from those reported in general hospitalised cohorts, and by contextualising two-year findings from mixed-severity populations using the CO-FLOW study as an illustrative example (Introduction, Background; Lines 80–82). These revisions aim to better align the cited literature with the pathophysiological context and recovery expectations specific to survivors of critical COVID-19.

Comment 2: There are several methodological choices deserve deeper discussion. The transition from predominantly face-to-face follow-up at 1 year to exclusively telephone-based follow-up at 3 years introduces a non-trivial risk of measurement drift, particularly for clinician-reported outcomes such as GOSE. Although the authors acknowledge this limitation later, it would strengthen the Methods section to explicitly justify why telephone administration is considered comparable, or to cite validation studies specific to ICU or post-COVID populations.

Response:

We thank the Reviewer for this important methodological comment. We have revised the Methods section to explicitly justify the use of telephone-based follow-up at three years, which was chosen as a pragmatic design decision to balance data completeness, feasibility, and participant burden, and to minimise loss to follow-up over the extended study period. While face-to-face follow-up may allow for richer clinical information, including contextual cues and involvement of next of kin, particularly relevant for clinician-reported outcomes such as the GOSE, the 3-year follow-up was deliberately designed around outcome instruments that do not require physical attendance. We further clarify that all GOSE assessments were conducted using a structured interview format and manualised scoring procedures developed to improve inter-rater reliability and ensure consistency across assessors and modes of administration. In addition, we now cite validation data demonstrating good agreement between telephone-based and face-to-face GOSE assessment (Methods; Lines 117–119, 140–145).

We also acknowledge in the Discussion that, although telephone-based administration of the GOSE has demonstrated good agreement with in-person assessment, face-to-face assessment may provide greater clinical depth, and formal validation specifically in post-COVID ICU populations remains limited. However, the consistency of longitudinal findings across multiple patient-reported outcome measures, which are less sensitive to mode of administration, supports the robustness of the observed trajectories despite the change in follow-up modality. Notably, all outcome measures except the GOSE were patient-reported (PROMs. These instruments demonstrated identical longitudinal trends between 1 and 3 years. This is now addressed as a study limitation (Discussion; Lines 462–466).

Comment 3: The absence of a formal sample size calculation is understandable given the population-based design, but the manuscript would benefit from a brief discussion of statistical power, especially for multivariable regression analyses. Readers are left uncertain whether null findings for ICU severity variables reflect true lack of association or limited power.

Response:

We thank the Reviewer for this important comment. We have now expanded the Strengths and Limitations section to explicitly address statistical power in relation to the multivariable regression analyses. We clarify that, although the overall sample size was sufficient for the primary longitudinal analyses, statistical power for multivariable regression analyses may have been limited. Accordingly, null findings, including the lack of an independent association for sex, should be interpreted with caution; however, effect estimates for ICU severity variables were generally close to null rather than highly imprecise, suggesting that the absence of association was not solely driven by limited power (Discussion; Strengths and limitations, Lines 467–471).

Comment 4: The regression modeling strategy is sound but constrained. By excluding variables with higher missingness and focusing primarily on baseline characteristics, the multivariable model is structurally predisposed to identify age and frailty while under-representing social, occupational, and rehabilitation-related determinants that may be more relevant at three years.

Response:

We thank the Reviewer for this thoughtful comment. We would like to clarify that several sociodemographic factors were included in the multivariable analyses but did not remain independently associated with incomplete recovery after multivariable adjustment. We have now clarified this interpretation in the Discussion, emphasising that, in this cohort, long-term outcomes at three years were more strongly shaped by individual vulnerability than by sociodemographic characteristics (Discussion; Lines 440–446).

Variables with substantial missingness were excluded to ensure model stability, and several clinically relevant variables could not be included simultaneously due to multicollinearity. These variables were therefore examined in univariable analyses and discussed accordingly. Rehabilitation-related factors at three years (such as fatigue, psychological symptoms, and mental HRQoL) that were assessed concurrently with functional outcome were primarily explored using correlation network analysis. This approach allowed us to characterise symptom clustering and interdependencies without implying causal direction. Rehabilitation-related aspects are addressed separately in the manuscript and discussed as part of the study’s limitations (Discussion; Strengths and limitations, Lines 478–479).

Comment 5: Results would benefit from a stronger emphasis on heterogeneity. While group-level changes are clear, readers are left without a sense of which subgroups are most affected beyond age and frailty. Stratified analyses—by age group, employment status, or baseline mental health—would add important clinical nuance and help clinicians identify patients at greatest risk.

Response:

We thank the Reviewer for this thoughtful and clinically relevant suggestion. Heterogeneity in long-term outcomes was explored using individual-level longitudinal analyses, multivariable regression, and correlation network modelling, which together allowed us to capture variation in recovery trajectories beyond group-level averages. To further address this point, we performed additional age-stratified analyses at the 3-year follow-up. These analyses supported our main finding that younger survivors had worse functional outcomes than older age groups, highlighting heterogeneity in recovery trajectories beyond frailty alone. The results are presented in S2 Table and integrated into the Results and Discussion (Lines 282–285, 422–423).

Age stratification was prioritised because age emerged as the strongest independent predictor in the multivariable model and was available without substantial missingness, whereas further stratification would have resulted in underpowered subgroups.

In addition, the manuscript already includes descriptive comparisons of participants with good versus incomplete recovery at 3 years, covering sociodemographic, occupational, and patient-reported characteristics. These descriptive results are presented to illustrate clinically relevant heterogeneity between outcome groups without implying statistical inference or causality.

Reviewer #2: The authors have done a substantial amount of work and the manuscript has improved. They have responded to my questions and comments satisfactorily.

Response:

We thank the Reviewer for their careful evaluation of the revised manuscript and for their positive feedback. We are pleased that the responses and revisions were satisfactory.

---

## [Editor Report · Decision Letter 2]

6 Jan 2026

Three-year functional, physical, and mental health outcomes after critical COVID-19: A prospective multicentre cohort study

PONE-D-25-38284R2

Dear Dr. Didriksson,

We’re pleased to inform you that your manuscript has been judged scientifically suitable for publication and will be formally accepted for publication once it meets all outstanding technical requirements.

Kind regards,

Andrea Martinuzzi

Academic Editor

PLOS One
---

## [Editor Report · Acceptance letter]

PONE-D-25-38284R2

PLOS One

Dear Dr. Didriksson,

I'm pleased to inform you that your manuscript has been deemed suitable for publication in PLOS One. Congratulations! Your manuscript is now being handed over to our production team.

Kind regards,

on behalf of

Dr. Andrea Martinuzzi

Academic Editor

PLOS One